# One Single Nucleotide Polymorphism of the *TRPM2* Channel Gene Identified as a Risk Factor in Bipolar Disorder Associates With Autism Spectrum Disorder in a Japanese Population

**DOI:** 10.3390/diseases8010004

**Published:** 2020-02-07

**Authors:** Naila Al Mahmuda, Shigeru Yokoyama, Toshio Munesue, Kenshi Hayashi, Kunimasa Yagi, Chiharu Tsuji, Haruhiro Higashida

**Affiliations:** 1Department of Basic Research on Social Recognition and Memory, Research Center for Child Mental Development, Kanazawa University, Kanazawa 920-8640, Japan; nlmahmuda@gmail.com (N.A.M.); shigeruy@med.kanazawa-u.ac.jp (S.Y.); munesue@med.kanazawa-u.ac.jp (T.M.); ctsuji@med.kanazawa-u.ac.jp (C.T.); 2Faculty of Business Administration, Eastern University, Dhaka 1205, Bangladesh; 3Division of Cardiovascular Medicine, Graduate School of Medical Science, Kanazawa University, Kanazawa 920-8641, Japan; kenshi@med.kanazawa-u.ac.jp (K.H.); yagikuni@icloud.com (K.Y.); 4Laboratory for Social Brain Studies, Research Institute of Molecular Medicine and Pathobiochemistry, Department of Biochemistry, Krasnoyarsk State Medical University named after Prof. V. F. Voino-Yasentsky, Krasnoyarsk 660022, Russia

**Keywords:** SNP, ASD, bipolar disorder, *TRPM2*, CD38

## Abstract

The transient receptor potential melastatin 2 (TRPM2) is a non-specific cation channel, resulting in Ca^2+^ influx at warm temperatures from 34 °C to 47 °C, thus including the body temperature range in mammals. TRPM2 channels are activated by β-NAD^+^, ADP-ribose (ADPR), cyclic ADPR, and 2′-deoxyadenosine 5′-diphosphoribose. It has been shown that TRPM2 cation channels and CD38, a type II or type III transmembrane protein with ADP-ribosyl cyclase activity, simultaneously play a role in heat-sensitive and NAD^+^ metabolite-dependent intracellular free Ca^2+^ concentration increases in hypothalamic oxytocinergic neurons. Subsequently, oxytocin (OT) is released to the brain. Impairment of OT release may induce social amnesia, one of the core symptoms of autism spectrum disorder (ASD). The risk of single nucleotide polymorphisms (SNPs) and variants of *TRPM2* have been reported in bipolar disorder, but not in ASD. Therefore, it is reasonable to examine whether SNPs or haplotypes in *TRPM2* are associated with ASD. Here, we report a case-control study with 147 ASD patients and 150 unselected volunteers at Kanazawa University Hospital in Japan. The sequence-specific primer-polymerase chain reaction method together with fluorescence correlation spectroscopy was applied. Of 14 SNPs examined, one SNP (rs933151) displayed a significant *p*-value (OR = 0.1798, 95% CI = 0.039, 0.83; Fisher’s exact test; *p* = 0.0196). The present research data suggest that rs93315, identified as a risk factor for bipolar disorder, is a possible association factor for ASD.

## 1. Introduction

Autism spectrum disorder (ASD) is a pervasive neurodevelopmental disease [1,2,3]. Although the etiology accounting for ASD is not totally clear [1,4], it is now understood that such neurodisorders are the result of a multitude of factors, including environmental factors and genetic susceptibility [5,6]. Many studies have led to the identification of candidate genes whose variants and single nucleotide polymorphisms (SNPs) might be associated with ASD [7,8]. A recent genetic analysis suggested that common and rare variants contribute to ASD by perturbation of wide neuronal, proteomic, and genetic networks [9,10,11]. Because of the high heritability, the majority of research on ASD has been focused on finding underlying genetic risks, with less emphasis on possible environmental factors [12].

The core symptom of ASD is social impairment with communication problems [13]. Many studies have suggested that oxytocin (OT), a hormone with nine amino acids, plays a critical role in social behavior in mammals and humans [14,15,16,17,18,19,20,21,22]. A single administration of OT was revealed to have beneficial effects on social and emotional processes or impairments in both healthy subjects and individuals with a variety of psychiatric diseases, including ASD [23,24,25,26,27,28]. More practically, repetitive administrations of OT into the nasal cavity of individuals with ASD results in the reduction of social behavioral impairment and increased episodes of social interaction in daily life [29,30,31,32,33,34,35,36,37,38,39,40,41,42,43,44,45,46,47,48,49,50,51,52,53,54,55,56]. To support OT accumulation in the brain after peripheral application, it has recently been reported that the receptor for advanced glycation end products (RAGE) recruits OT from the blood circulation crossing over the blood–brain barrier, because RAGE is an OT-binding partner and carrier [57,58].

OT is released into the brain and is essential for social behavioral actions [15,18,22,24,28,59,60,61]. Recent studies revealed that more OT is released in a hypothalamus culture isolated from subordinate mice in group-housed males than from dominant mice, especially after cage-switch stress [62]. Upon stimulation with cyclic ADP-ribose (cADPR) through push-pull canulae, OT concentrations in micro-perfusates at the paraventricular nucleus were higher in subordinate mice compared with ordinate mice. The OT concentration of cerebrospinal fluid (CSF) was enhanced in endotoxin-shock mice with fever in comparison to controls without hyperthermia [62,63]. In mice exposed to the new environmental stress, the CSF OT level transiently increased 5 min after the start of exposure, during which the rectal temperature also increased. OT release under such conditions was sensitive to mRNA levels of the transient receptor potential melastatin 2 (TRPM2, previously known as TRPC7 or LTRPC2TRPM2) in the hypothalamus and to a TRPM2 antagonist [62]. Because TRPM2 cation channels are warm-sensitive channels [64,65], the above findings indicate that hyperthermia regulates hypothalamic OT secretion during social stress by intracellular free Ca^2+^ concentration elevation resulting from TRPM2-dependent Ca^2+^ influx and CD38-controled Ca^2+^ mobilization.

Therefore, it is rational to consider that the *TRPM2* gene may contribute to the etiology of ASD and that the TRPM2 channel is a potential therapeutic target in social impairments in ASD with respect to OT release. Interestingly, abnormal functions of TRPM2 have been reported to be risk factors in bipolar disorder, although there are no genetic studies on *TRPM2* for ASD [66]. To assess this, we screened 14 SNPs in the *TRPM2* gene and verified the association of SNPs of *TRPM2* with ASD in the same Japanese population, in which genetic variants and/or SNPs of *CD38, BST1/CD157*, and *SLC19A1* were reported as risk factors of ASD [29,67,68]. Finally, we will discuss pathophysiological roles of TRPM2 channels in the brain and neurological diseases.

## 2. Experimental Section

### 2.1. Subjects/Children with ASD and Healthy Controls

There were 147 ASD subjects, including 113 males and 34 females (mean age: 15.6 ± 0.6 years) from the psychiatry and child mental clinics of Kanazawa University Hospital [29,68]. All patients satisfied the Diagnostic and Statistical Manual of Mental Disorders (DSM)-IV criteria for pervasive developmental disorder. Such subjects were diagnosed by psychiatrists through interviews and observations of their behaviors in the playroom [29]. They had no history of noticeable physical abnormalities. The diagnosis of ASD for all participants was independently confirmed by an experienced child psychiatrist by semi-structured behavior observations as well as through interviews with the patients and their mothers and/or fathers. In the process of evaluating ASD during interviews, the examiner used one of the following methods: the Asperger Syndrome Diagnostic Interview, the Autism Diagnostic Interview-Revised, the Pervasive Developmental Disorders Autism Society Japan Rating Scale, the Diagnostic Interview for Social and Communication Disorders, or the Tokyo Autistic Behavior Scale [46]. We excluded two ASD subjects due to a lack of results in the SNP analysis.

As the control group, 115 male and 35 female volunteers (mean age: 23.8 ± 0.3 years) who had no history of physical or mental disorders were accounted. However, four control subjects were excluded due to insufficient results in the SNP analysis. All participants were Japanese, with Japanese parents and grandparents. This study was approved by the ethics committees of Kanazawa University School of Medicine. All examinations were performed after informed consent was obtained from the parents or guardians of all children in accordance with the Declaration of Helsinki. 

### 2.2. Genotyping

Genomic DNA was extracted as previously described [69] from venous blood samples using a kit (Wizard Genomic DNA Purification kit; Promega, Madison, WI, USA), or from nails using the ISOHAIR DNA extraction kit (Nippon Gene, Tokyo, Japan). In some cases, the whole genome amplification method (the REPLI-g kit; Qiagen, Hilden, Germany) was applied in order to increase DNA concentrations. The genotypes of each SNP were determined at Kurabo Industries Ltd. (Osaka, Japan) by the sequence-specific primer (SSP)-PCR method, which was combined with fluorescence correlation spectroscopy [70]. We picked up tagging SNPs that cover common variations. Linkage disequilibrium (LD) structures over the *TRPM2* gene were analyzed by the Tagger program equipped with the Haploview v4.2 software (Broad Institute of MIT and Harvard, Cambridge, MA, USA). We used the dbSNP database for tagging SNPs [71]: the HapMap genome browser, release 27 (the National Institutes of Health, Bethesda, MD, USA), the JPT (Japanese individuals from Tokyo, Japan), CHB (Han Chinese individuals from Beijing, China), ASW (African ancestry in Southwest USA), and CEU (Utah residents of northern and western European ancestry). The selection of tagging SNPs was based on pairwise tagging. The minor allele frequency ≥5% was selected in any one of the different ethnicities. 

### 2.3. Statistical Analysis

A contingency table and Fisher’s exact test (GraphPad Prism 6; GraphPad Software, San Diego, CA, USA) were used to examine genotypes and allele frequencies. In this study, *p* < 0.05 was taken to indicate statistical significance. The genotype frequency distributions calculated were compared with those expected from the Hardy–Weinberg equilibrium. The chi “χ” squared test was applied. Statistical power was processed using the Genetic Power Calculator [72,73,74,75]. Calculations were performed by estimating a population prevalence of 0.015 for ASD [74] and a *D*’ value of 1 between the marker and disease, with a false positive rate of 5%.

## 3. Results

In this study, 16 SNPs of *TRPM2* on human chromosome 21, whose physical locations are shown in Figure 1, were subjected to analysis in case–control association samples of the initial samples of 145 cases and 146 controls. First, two SNPs (rs3788122 and rs9982220) were excluded because of insufficient genotyping data. In 13 SNPs, an association with ASD was not found. One SNP, rs933151, provided evidence for an association. rs933151 was significant, with the empirical *p*-value of 0.0196 (Table 1). However, rs933151 did not show evidence of an association, with χ^2^ = 1.940 (*p* = 0.3791). This indicates that rs933151 alone or in combination with others is a weak risk factor for ASD. Interestingly, it has been reported that rs933151 is associated with bipolar disorder [76]. This SNP was further analyzed.

Tests of Hardy–Weinberg equilibrium deviations were performed for each marker in case and control individuals. Polymorphisms showed evidence of deviation from the Hardy–Weinberg equilibrium. The genotyping rate was above 95%. LD analysis of these SNPs identified three haplotype blocks, one of which (Block 1; Figure 2) consisted of three SNPs, including the one (rs933151) with the lowest *p*-value among those analyzed: Statistic χ^2^ = 4.550 (*p* = 0.1028).

Hardy—Weinberg analysis showed a significant frequency variation in the A/A, A/G, and G/G genotypes of rs933151 of *TRPM2* in 145 cases and 146 controls (Appendix A). The frequency of the A/A, A/G, and G/G genotypes was 0.64, 0.32, and 0.04 in the control subjects, respectively, and 0.59, 0.36, and 0.05 in ASD subjects, respectively. Chi square (χ^2^) goodness-of-fit test values for the controls and ASD subjects were 4.13 (*p* = 0.0421) and 1.94 (*p* = 0.1637), respectively. In the A/A and A/G genotypes, allele frequency was found to be significant at the 0.05 level. Furthermore, the Elston–Forthofer average test revealed A = 445 and 212 for the control and ASD subjects, respectively, confirming the significant differences in the allele frequency of rs933151.

## 4. Discussion

Specific SNPs or variants associated with multiple neuronal disorders and cross-disorder phenotypes have been identified [77]. The major finding of this study is the weak association of ASD with the SNP rs933151 in intron 20 of *TRPM2* (Figure 1) [76,78,79,80]. Similarly, with respect to *TRPM2*, it has been reported that the SNP rs1556314 in exon 11 was associated with bipolar disorder type I in the Caucasian case control dataset and in the family design. In addition, the C-T-A haplotype of SNPs rs1618355, rs933151, and rs749909 of the *TRPM2* gene was significantly associated with early age at onset in bipolar disorder type I families [76]. An association between ASD or attention-deficit hyperactivity disorder and *TRPM2* has not been reported yet [81]. Thus, the current article reveals for the first time that rs933151 may be related to both ASD and bipolar disorder. This finding regarding rs933151 seems to be particularly meaningful in the psychiatric field, because ASD sometimes occurs concurrently with bipolar disease [1]. However, we should keep in mind that the insensitivity of this study design of population stratification creates a factor that can lead to false association [82].

In relation to hyperthermia involving TRPM2 channels, a beneficial effect of fever in ASD patients has been reported [62]. Some ASD children exhibit improvements in their behavioral characteristics during febrile incidents [62,63,83,84,85,86,87]. Febrigenesis and behavioral changes induced by fever in ASD may depend on the facilitation of brain OT release or the normalization of the impaired locus coeruleus-noradrenergic system [87].

Zhong et al. [62] and Higashida et al. [63] reported the new neuroendocrinological evidence for involvement of the warm temperature-sensor cation channel TRPM2 in OT release. They showed that incubation with cADPR induced minor but sustained increases in the OT concentration in the culture medium of the hypothalamus. However, the OT concentration markedly increased (by approximately 4-fold) with simultaneous heat stimulation from 35 °C to 38.5 °C compared with the pre-stimulation level [62]. The mechanism for enhanced OT release is owing to increased cellular Ca^2+^ concentrations. It is speculated that cADPR activates Ca^2+^ mobilization from ryanodine-sensitive Ca^2+^ pools [15] and enhances Ca^2+^ influx by TRPM2 channels [62], because cADPR binds to the NUDT9-H domain of TRPM2 channels for opening [88]. Of course, heat gates TRPM2 Ca^2+^ channels.

The same mechanism by TRPM2 would be expected in ASD patients. Hyperthermia may induce temporal remission, probably because of the increased release of OT in the brain in ASD subjects. Thus, investigating how the SNP rs933151 and phosphorylated glycogen synthase kinase-3 [66,89] may influence this TRPM2-sensitive OT release is the next step we have to take.

Recently, the physiological functions of TRPM2 in the brain have been identified [66,90]. TRPM2 resides not only in neurons, but also astrocytes or microglial cells, which play a role in cognitive functions [91,92,93,94]. The mRNA expression profile in microglia of TRPM2 in the hypothalamus and striatum of the developing rat brain has been shown to be paralleled with the perinatal expression timeline for microglial infiltration and maturation [95]. TRPM2 has been shown to be involved in embryonic neurogenesis [96]. Moreover, aberrant TRPM2 function seems to be associated with psychiatric and neurological diseases, such as stress-induced depression [97], epilepsy [94], ischemic brain injury [98,99,100], and glioma invasion [99]. Therefore, it is possible that TRPM2 mutants and TRPM2 expression during neuronal development may contribute in the pathogenesis of ASD [1,3,7,96] and bipolar disorder [66,89,90,91,96,101].

## 5. Conclusions

Although limitations in the small size of the current cohort are clear, we reported a weak association between ASD and the SNP (rs933151) of the *TRPM2* gene in a Japanese population. The rs933151 SNP has already been reported as a risk factor in bipolar disorder. Our results necessitate further analysis of *TRPM2* SNPs in ASD patients on a large scale. It remains unclear whether the variants causing the amino acid mutation of *TRPM2* found in bipolar disorder patients [66,76,80,89] are also risk factors in ASD.

## Figures and Tables

**Figure 1 diseases-08-00004-f001:**
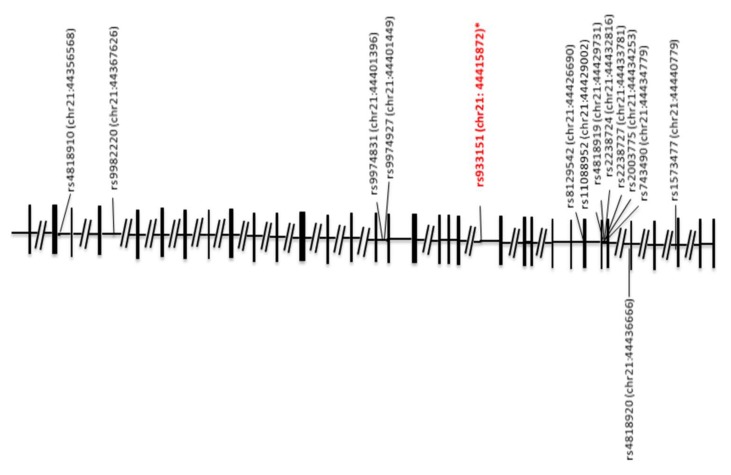
Schematic genomic structure of the human *TRPM2* gene and locations of single nucleotide polymorphisms (SNPs). The exon-intron organization is depicted based on GenBank accession numbers NC_000021. The SNP (red and italicized) represents a significant association with ASD in allele and/or genotype frequencies in the present study; the asterisk indicates a previously reported bipolar disorder association [76]. The locations of the SNPs on human chromosome 21 (chr21) are indicated in parentheses; numbers after colons represent genomic positions based on the human genome assembly GRCh38/hg19 at the University of California Santa Cruz (UCSC) Genome Bioinformatics Site.

**Figure 2 diseases-08-00004-f002:**
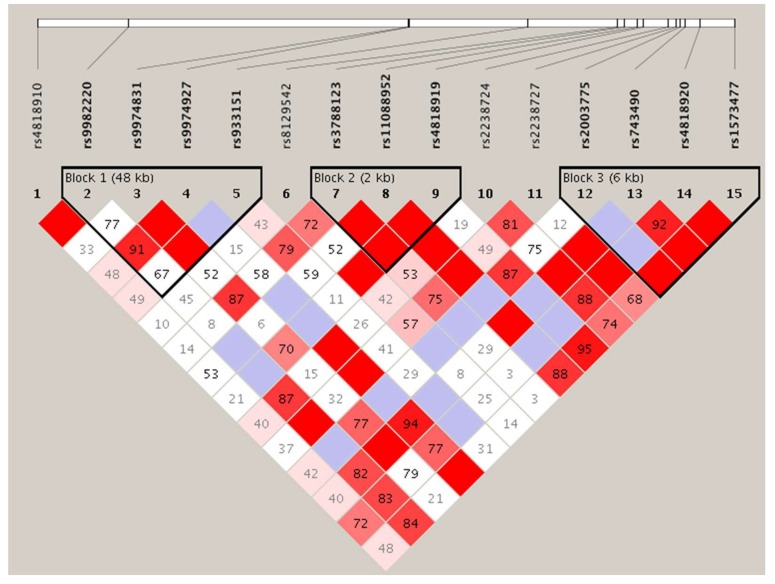
Linkage disequilibrium plot of SNPs of *TRPM2* in the samples examined. Numbers in squares indicate *D*′ values. The blocks are defined following the four-gamete rule [75] Explanation of color scheme: If *D*′ < 1 and LOD (log of the likelihood odds ratio) <2, the cell color is white; if *D*′ = 1 and LOD < 2, it is blue; if *D*′ < 1 and LOD ≥ 2, shades of pink/red; if *D*′ = 1 and LOD ≥ 2, bright red.

**Table 1 diseases-08-00004-t001:** Genotype and allele frequencies of rs933151 in control and ASD subjects.

	Cases	Controls	Odds Ratio	*P*
			(95% CI)	
Genotype	(*n* = 145)	(*n* = 146)		
A/A	88 (60.7%)	89 (61.0%)	Referent	
A/G	46 (31.7%)	55 (38.2%)	1.182 (0.72, 1.93)	0.5341
G/G	11 (7.6%)	2 (1.2%)	0.1798 (**0.039**, 0.83)	**0.0196**
Allele	(*n* = 290)	(*n* = 292)		
A	222 (76.6%)	233 (79.8%)	Referent	
G	68 (23.4%)	59 (20.2%)	0.8267 (0.56, 1.23)	0.3672

CI: confidence interval. *p*-values obtained by the Fisher exact test for genotype and by Chi-square test for allele are given. Bold means significance.

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
