# Peer review of "One Single Nucleotide Polymorphism of the TRPM2 Channel Gene Identified as a Risk Factor in Bipolar Disorder Associates With Autism Spectrum Disorder in a Japanese Population"

_diseases, 2020, doi:10.3390/diseases8010004_

Round 1
Reviewer 1 Report
This work reported weak association between ASD and the SNP (rs933151) of the TRPM2 gene, which has already been reported as a risk factor in bipolar disorder, in a Japanese population. Despite of the small sample size, the study was performed in a clear manner and the manuscript was well written.
A couple minor concerns:
1. A few citations were marked in red (Line 142, 149, 157, 163) but not showing specific meanings.
2. Table 1: was the statistics corrected with multiple comparison?
Author Response
A few citations were marked in red (Line 142, 149, 157, 166) but not showing specific meanings.
Thank you very much. We carefully evaluated these references, and corrected.
Table 1: was the statistics corrected with multiple comparison?We appreciated the comment. The statistics was corrected with multiple comparison, but no significance result was observed after correction, and hence we stated that the ASP is a weak risk factor for ASD. Table 1 was amended after careful inspection of our result again.
Reviewer 2 Report
In this report Al Mahmuda et al evaluated a possible association between onTRPM2 gene polymorphism and ASD in a Japanese population cohort of ASD children and healthy controls. Among the 14 SNPs analysed only the rs933151 was reported to be associated with ASD.
I have many concern both regarding data evaluation and soundness of the study.
Results are poorly described and data reported in table 1 may be checked because there is discrepancy between genotype distribution and number of case and controls. Since the only result of this study is based in this table, the significance of the study fail.
Moreover the author cite Belrose et al and Jang Y et al studies of association of TRPM2 SNPs with bipolar disorder assessing to confirm rs933151 association, but indeed no mention regarding this specific SNP was reported in these studies. The only association of rs933151 with bipolar disorder was hypothesized by XU C et al. 2009, as rs1618355, rs933151, and rs749909 haplotype association with early age at onset of bipolar disorders. It was not clear if the authors did evaluate any haplotype comparison between cases and controls.
Author Response
For comments by eviewer #2:
Results are poorly described and data reported in table 1 may be checked because there is discrepancy between genotype distribution and number of case and controls. Since the only result of this study is based in this table, the significance of the study fail.
Thank you very much for your valuable comment. This has definitely helped us to improve the quality of the Table 1. The new table 1 has been checked and corrected in red.
Moreover the author cite Belrose et al and Jang Y et al studies of association of TRPM2 SNPs with bipolar disorder assessing to confirm rs933151 association, but indeed no mention regarding this specific SNP was reported in these studies. The only association of rs933151 with bipolar disorder was hypothesized by XU C et al. 2009, as rs1618355, rs933151, and rs749909 haplotype association with early age at onset of bipolar disorders. It was not clear if the authors did evaluate any haplotype comparison between cases and controls.
We appreciated these scientific comments. The data allowed us to evaluate the approaches in scenarios, where the causal variants are individual SNPs, rather than haplotypes. No haplotype comparison between cases and controls were evaluated. Therefore, we stopped to discuss more.
Reviewer 3 Report
In this study, the authors demonstrated the weak association between autism spectrum disorder (ASD) and the SNP (rs933151) of the TRPM2 gene. I reviewed the manuscript carefully. I feel … the examinations were performed well, the findings are clear and interesting, and the conclusion is reasonable. So, the current evidence will be useful for better understanding of ASD etiology and may contribute to the development of novel ASD drug therapy. However, I have a bit concern as follows.
1) Introduction & Discussion: I recommend you to add more information of the pathophysiological roles of TRPM2 in psychiatric function (bipolar disorder pathology, if possible) in the Introduction. In addition, I ask you to describe the possible involvement of TRPM2 in ASD etiology in the Discussion. To do so, I suggest to you to describe the brain distribution of TRPM2 including neurodevelopmental alteration of TRPM2 expression in the Introduction and/or Discussion.
2) Sex difference: The male:female ratio of ASD is generally considered to be 4:1. In this study, you analyzed data without dividing males and females. Is there any information for sex difference in the association between ASD and the SNP (rs933151) of the TRPM2 gene? If you have such information, please add it in the revised version.
Author Response
Reviewer #3:
1) Introduction & Discussion: I recommend you to add more information of the pathophysiological roles of TRPM2 in psychiatric function (bipolar disorder pathology, if possible) in the Introduction. In addition, I ask you to describe the possible involvement of TRPM2 in ASD etiology in the Discussion. To do so, I suggest to you to describe the brain distribution of TRPM2 including neurodevelopmental alteration of TRPM2 expression in the Introduction and/or Discussion.
Thank you very much for these thoughtful comments. Since there are few reports on TRPM2 and ASD, we barely cite references on this point. However, we stated for discussion in Introduction and discussed at the end of the discussion section on the pathophysiological roles of TRPM2 in psychiatric disease. I found some very interesting reports have reported. Thus, new 12 such references (from 90-101) were added.
2) Sex difference: The male:female ratio of ASD is generally considered to be 4:1. In this study, you analyzed data without dividing males and females. Is there any information for sex difference in the association between ASD and the SNP (rs933151) of the TRPM2 gene? If you have such information, please add it in the revised version.
Thank you very much for your valuable concern. we will definitely consider this point in our next step of research on the TRPM2 gene. In this study, sex difference was not considered in the association between ASD and the SNP (rs933151) of the TRPM2 gene. To our knowledge, there is no such report.
Round 2
Reviewer 2 Report
I appreciate that tha authors corrected the numebrs in table 1 but still there are discrepancies between number of case = 147( 113 males and 34 females) and controls =150 (115 male and 35 female) reported in the text and the ones described in table 1. (145 and 146 respectively) . The authors may justify this discrepancy.
Moreover I suggested a more detailed description of the results, in results paragraph, but this point was not assessed by the authors.
Finally I didn’t understand why the authors didn’t evaluate haplotype comparison If they have raw data for each SNP they can use bioinformatic tools available on web or ask to a statistician . It would be an improving point for the manuscript both if the results would be negative or positive.
Author Response
I appreciate that authors corrected the numebrs in table 1 but still there are discrepancies between number of case = 147( 113 males and 34 females) and controls =150 (115 male and 35 female) reported in the text and the ones described in table 1. (145 and 146 respectively) . The authors may justify this discrepancy.
Thank you for your careful comment. SNPs of 2 cases and 4 controls were not detected. That is why 2 of cases and 4 of controls were excluded from the experiment. Therefore we explained it in the text as follow:
We excluded 2 ASD subjects, because of no result in SNP analysis. As the control group, 115 male and 35 female volunteers (with the mean age of 23.8 ± 0.3 years) who had no history of physical or mental disorders were accounted. However, 4 control subjects were excluded by insufficient results in SNP analysis.
Moreover I suggested a more detailed description of the results, in results paragraph, but this point was not assessed by the authors.
We are sorry for no correction for the last version. According to your kind suggestion, we amended the result section as follow:
ResultsIn this study, 16 SNPs of TRPM2 on human chromosome 21, whose physical locations are shown in Figure 1, were subjected to analysis in case–control association samples of the initial samples of 145 cases and 146 controls. First, two SNPs (rs3788122 and rs9982220) were excluded, because of insufficient genotyping data. In 13 SNPs, association with ASD was not found. One SNP rs933151 provided evidence for association. rs933151 was significant with the empirical P-value of 0.0196 (Table 1). However, rs933151 did not show evidence for association with statistic c2 = 1.940 (P = 0.3791). This indicates that rs933151 alone or in combination with others is a weak risk factor for ASD. Interestingly, it has been reported that rs933151 is associated with bipolar disorder [76], this SNP was further analyzed.
Tests of Hardy-Weinberg equilibrium deviations were performed for each marker in case and control individuals. Polymorphisms showed evidence of deviation from the Hardy-Weinberg equilibrium. The genotyping rate was above 95%. LD analysis of these SNPs identified three haplotype blocks, one of which (Block 1; Figure 2) consisted of three SNPs including the one (rs933151) with the lowest p-value among those analyzed: Statistic c2 = 4.550 (P = 0.1028).
Hardy-Weinberger analyse showed a significant frequency variation in the A/A, A/G, and G/G genotypes of rs933151 of TRPM2 in 145 cases and 146 controls (Supplementary Table 1). The frequency of the A/A, A/G, and G/G genotypes was 0.64, 0.32, and 0.04 in the control subjects, and 0.59, 0.36, and 0.05 in ASD subjects, respectively. Chi square (c2) goodness-of-fit test values for the controls and ASD subjects were 4.13 (P = 0.0421) and 1.94 (P = 0.1637), respectively. In the A/A and A/G allele frequency was found to be significant at the 0.05-levels. Furthermore, the Elston-Forthofer average test revealed A = 445 and 212 for the control and ASD subjects, respectively, confirming the significant differences in the allele frequency of rs933151.
Finally I didn’t understand why the authors didn’t evaluate haplotype comparison If they have raw data for each SNP they can use bioinformatic tools available on web or ask to a statistician . It would be an improving point for the manuscript both if the results would be negative or positive.
Thank you very much for your valuable suggestions. In view of this, to test the assumption of Hardy-Weinberg Equilibrium (HWE) values, the standard chi-squared statistics on 1 degree of freedom and a corresponding p-value were also generated for this variant using GraphPad Prism 6. Data has been analyzed by comparing observed distribution with expected and chi-square test for goodness of fit. These results are shown in new Supplementary Table 1 and described in the Results section of the revised version.
